# IMPROVING THE STRENGTH OF HUMAN-LIKE MODELS IN CHESS

## ABSTRACT

Designing AI systems that capture human-like behavior has attracted growing attention in applications where humans may want to learn from, or need to collaborate with, these AI systems. Many existing works in designing human-like AI have taken a supervised learning approach that learns from data of human behavior, with the goal of creating models that can accurately predict human behavior. While this approach has shown success in capturing human behavior at different skill levels and even identifying individual behavioral styles, it also suffers from the drawback of mimicking human mistakes. Moreover, existing models only capture a snapshot of human behavior, leaving the question of how to improve them—e.g., from one human skill level to a stronger one—largely unanswered. Using chess as an experimental domain, we investigate the question of teaching an existing human-like model to be stronger using a data-efficient curriculum, while maintaining the model's human similarity. To achieve this goal, we extend the concept of curriculum learning to settings with multiple labeling strategies, allowing us to vary both the *curriculum* (dataset) and the *teacher* (labeling strategy). We find that the choice of teacher has a strong impact on both playing strength and human similarity; for example, a teacher that is too strong can be less effective at improving playing strength and degrade human similarity more rapidly. We also find that the choice of curriculum can impact these metrics, but to a smaller extent; for example, training on a curriculum of human mistakes provides only a marginal benefit over training on a random curriculum. Finally, we show that our strengthened models achieve human similarity on datasets corresponding to their strengthened level of play, suggesting that our curriculum training methodology is improving them in human-like steps.

## 1 INTRODUCTION

AI systems are growing increasingly capable at solving tasks, making decisions, and assisting humans in a wide number of domains. In games such as Chess, Go, and Poker, AI has demonstrated clear superiority over human performance (Silver et al., 2018; Brown & Sandholm, 2019). In domains such as marketing, transportation, medicine, law, hiring, and finance, AI decision making is capable enough to be deployed alongside humans in many real-life scenarios, and in some cases, AI performance is sufficient to take over entirely.

More recently, researchers have looked into developing AI with the explicit goal of replicating human decision making, in contrast to simply optimizing for AI performance. There are several reasons for doing this. In domains such as autonomous driving, developing human-like AI can result in increased safety for both humans and AI, by providing an understanding of human driving tendencies (Hecker et al., 2019). In situations where AI is assisting or educating humans, humans exhibit a higher level of trust when the AI is human-like (Wang et al., 2019; Li et al., 2021; Kim et al., 2022), and higher levels of satisfaction when interacting with human-like AI (Amigó et al., 2006; Ragot et al., 2020; Jenneboer et al., 2022). In gaming, human-like AI can offer opportunities to practice against models of real opponents (McIlroy-Young et al., 2022b;a), and is generally more enjoyable to play with in competitive or cooperative play (Zhang et al., 2021; Soni & Hingston, 2008).

Modern efforts to create human-like AI are improving rapidly, but these systems naturally lead to reduced task performance as a result of focusing on human similarity. For example in chess,

researchers have used a supervised learning approach on human game data to create models that match human decision-making at different skill levels (McIlroy-Young et al., 2020); these models can even be fine-tuned to match the decisions of an individual player and distinguish between individual playing styles (McIlroy-Young et al., 2022b; 2021). Since these models strive to match human behavior at different skill levels, they also adopt the human tendencies and mistakes made at those levels. Moreover, each of these models only represents a snapshot of human behavior, with no investigation of how to improve the models, e.g., from one human skill level to a stronger one.

In this paper, we attempt to address these questions by developing a methodology for improving the strength of human-like models, while retaining their human similarity. Specifically, given a human-like model at a particular skill level, our goal is to improve the model's strength in a data-efficient manner, without degrading its human similarity by too much. Thus, we evaluate these human-like models along a trade-off of three metrics: strength, data efficiency, and human similarity (§3.2).

Chess is an ideal domain for our study due to the availability of AI systems at both ends of the strength vs. human similarity spectrum. Chess engines like Stockfish and Leela (the open-source replication of AlphaZero (Silver et al., 2018)) dominate the best human chess players in strength, while Maia Chess (McIlroy-Young et al., 2020) provides a framework for training models that can accurately mimic human behavior at different skill levels. Additionally, using chess affords us a massive amount of public human data, which can be used for training or further analysis. Finally, chess has a well-defined ruleset, which allows us to evaluate the playing strength of any chess model we obtain using computationally cheap simulations.

In devising a methodology for strengthening human-like models, we considered two main approaches. The first is to leverage a state-of-the-art chess learning framework, such as the deep reinforcement learning framework of AlphaZero, but with an additional constraint to match human moves. Such a constrained optimization approach, however, is difficult to integrate into existing human-like models. Additionally, this process is far removed from any real human learning process. Instead, we chose a methodology that more closely mimics how a human might learn: starting with an existing human-like model, which we call the *student*, we fine-tune the model using a customized training curriculum. This approach gives us control over how we trade off human similarity for playing strength, while also developing training curricula that are interpretable to humans. We implement this methodology by replicating and extending Maia's framework for training human-like models and fine-tuning them on individual player data (McIlroy-Young et al., 2022b), except we train our own models and fine-tune them on different training curricula that we design.

The core idea behind our approach is quite similar to the concept of Curriculum Learning developed by Bengio et al. (2009), which looks at selecting the most efficient ordering of the data for training a model. However, we extend this concept in an important way, by not just looking at the selection of the data, but also considering the *strategy for labeling the data*. That is, we distinguish between the input dataset, which we call the *curriculum*, and the labeling strategy, which we call the *teacher*. In the case of chess, the curriculum corresponds to a set of chess positions, and the teacher could be any chess model that suggests which move to make in each position. Whereas traditional curriculum learning assumes a single, "correct" label, in our work we consider chess engines and human-like models of varying strength, which may have different opinions about which move to make in a given position. In our evaluation, we explore what happens when we vary both the curriculum and the teacher (§4.1), vary only the teacher (§4.2), or vary only the curriculum (§4.3).

Our results lead to several interesting conclusions. We find that the choice of teacher has a strong impact on both playing strength and human similarity, and it is not the case that the strongest teacher is the best choice. For example, a teacher like Stockfish is too strong and different when the student is a human-like model of novice strength. Instead, using a stronger human-like model as a teacher leads to much better outcomes in strength, human similarity, and training efficiency. We also find that the choice of curriculum can impact these metrics, but to a smaller extent. For example, training on a curriculum of positions the student made mistakes on provides only a marginal benefit over training on random positions the student encountered. Finally, we show that our strengthened models achieve human similarity on datasets corresponding to their strengthened level of play (§4.4), suggesting that our curriculum training methodology is improving these models in human-like steps.

## 2   RELATED WORK

Chess has been a useful domain for measuring the capability of AI, and researchers for half a century have worked on developing stronger chess engines. The first breakthrough occurred in 1997, when a chess supercomputer, Deep Blue, defeated the reigning world chess champion in a match (Campbell et al., 2002). By the mid-2000s, chess engines based on alpha-beta game tree search could consistently beat the strongest humans on standard computer hardware, and this trend continues to the current day with Stockfish, currently the strongest chess engine in the world.

With the announcement of AlphaZero, Silver et al. (2018) showed that chess engines trained using deep reinforcement learning could rival traditional alpha-beta engines. Recently, McIlroy-Young et al. (2020) released the Maia Chess engine, which repurposes the AlphaZero architecture to use supervised learning on human games to mimic human play directly, instead of trying to learn optimal play. Maia was trained on a massive public human chess dataset provided by the website Lichess.org, and showed the ability to mimic human behavior at a variety of different skill levels in chess. Further work by McIlroy-Young et al. (2022b) expanded the capability of Maia to mimic individual chess players as well as distinguish between individual playing styles (McIlroy-Young et al., 2021). Overall, the domain of chess contains some of the most sophisticated work on human-like AI, further supporting its position as the "Drosophila of AI" (McCarthy, 1990).

However, research on human-like AI extends far beyond chess. Jacob et al. (2022) have developed human-like AI for games like Go and Diplomacy. Hecker et al. (2019) show that autonomous vehicles can be trained to drive in a human-like manner using a quantitative measure of human similarity. Hu & Dong (2021) investigate the problem of making existing object detection algorithms more human-like, in order to pass Turing tests and defeat adversarial attacks.

In general, optimizing AI towards human similarity will necessarily lead to reduced task performance, especially in scenarios where (non-human-like) AI achieves superhuman performance. In this paper, we choose to handle this trade-off by extending the Curriculum Learning concept of Bengio et al. (2009) to support varying both the curriculum and the labeling strategy (teacher). When training an existing human-like model on a particular curriculum and teacher, we adopt the fine-tuning methodology of McIlroy-Young et al. (2022b), which itself draws on recent methods from the transfer learning literature (Sun et al., 2019; Yosinski et al., 2014; Kornblith et al., 2019)

While we intend curriculum training to be a domain-agnostic framework for efficiently developing strong and human-like AI systems, our choice of chess allows us to draw from existing chess pedagogical literature to inform our choices of specific curricula to investigate. There is strong empirical evidence supporting the notion that human chess training can build knowledge to improve future skill (Vaci et al., 2019). There are many avenues that human chess players can take to study, e.g., openings, endgames, tactics, game analysis, etc. However, there is a wide disagreement as to which of these, if any, is the most optimal path for training (Gobet & Jansen, 2006).

## 3   METHODOLOGY

The goal of this work is to investigate data-efficient approaches for improving the strength of a given human-like model, while maintaining its human-like property. In this section, we first describe the procedure we use to train human-like models in the domain of chess. We then define our performance metrics, which allow us to evaluating the trade-off between data efficiency, model strength, and human similarity when improving a given model. Finally, we introduce our curriculum training approach, which extends the concept of curriculum learning to multiple teachers, enabling us to examine the impact of varying both the curriculum (dataset) and the teaching advice (label strategy).

### 3.1   TRAINING HUMAN-LIKE MODELS IN CHESS

To develop human-like models, we replicate and extend the Maia framework developed by McIlroy-Young et al. (2020). We acquire a database of human chess moves from the online platform Lichess.org, where each move is labeled according to the player's Elo (skill) rating. We then group the moves by Elo rating, and use supervised learning to train a neural network to predict what move a human would play in a given chess position. For example, we train a model on moves made by players rated between 1150 to 1249, and we denote this model as Maia-1200. The neural network

we use is a slightly modified version of Maia's neural network, which itself is adapted from Leela's architecture. Notably, we significantly optimized the preprocessing pipeline and trained a broader set of models, going up to Maia-2600. Further information about our training procedure can be found in the appendix.

## 3.2 METRICS

Given a human-like model, we aim to improve its strength in a data-efficient manner, while maintaining its human similarity. We discuss the metrics we use to evaluate these performance measures.

**Data efficiency: Data size.** For data efficiency, we measure the amount of data used to train or update the model. Note that while we have access to an abundant amount of data in the domain of chess, if we are to generalize our research to other domains with practical data limitations, minimizing the amount of data usage will play an important role (§5).

**Human similarity: Move-matching accuracy.** To evaluate whether the model maintains its human-like playing style, we evaluate how well the updated model can predict human behavioral data. In general, we perform this evaluation specifically on the behavioral data used to train the original model, as this provides a direct way to measure the deviation from the original model. However, in the case of chess, we also have access to human data at a broad range of skill levels, so we also present results to evaluate our updated models against this additional data (§4.4).

To provide more insight into this metric, we trained a set of human-like models (following the procedure in §3.1) from Maia-1200 to Maia-2600, and measure their move-matching accuracy on the datasets with player ratings from 1200 to 2600. The results, shown in Figure 1a, demonstrate that the Maia-$K$ model is the most accurate in predicting the data of players of rating $K$. In particular, it is not the case that higher-rated models are better at predicting lower-rated datasets. This result replicates the findings of McIlroy-Young et al. (2020) and provides evidence that move-matching accuracy is a reasonable measure to evaluate whether the updated model is still human-like.

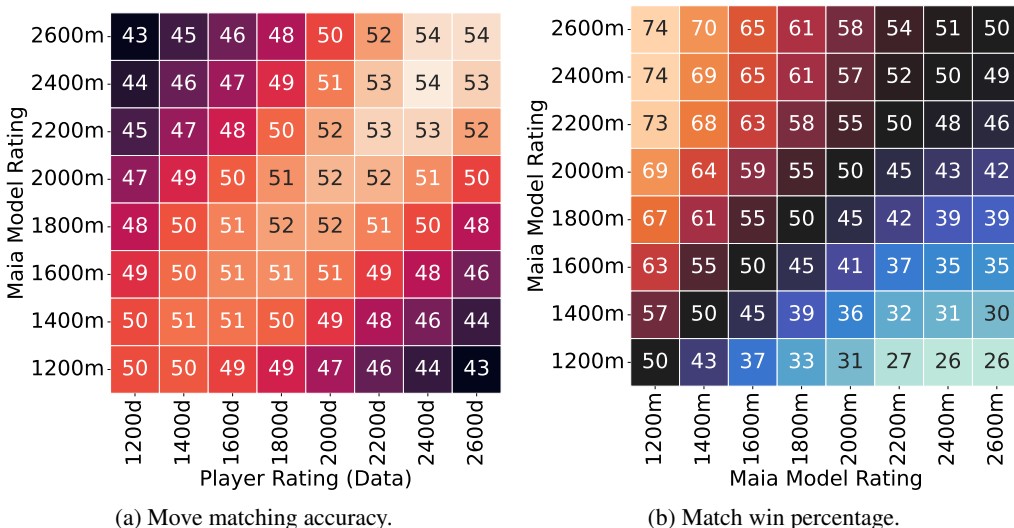

(a) Move matching accuracy.          (b) Match win percentage.

Figure 1: Illustrating the metrics of move-matching accuracy and match win percentage.

**Strength: Match win percentage.** To evaluate the strength of the updated model, we have the updated model play against the original model in a large number of full-length chess games (until checkmate or a draw is reached), and measure the match win percentage of the updated model. This win percentage is then used as a proxy for measuring the model strength.

To provide more insight into this metric, we have the models from Maia-1200 to Maia-2600 play against each other. The match win percentages (based on 5120 games played between each pair of models) are shown in Figure 1b. We can see that as we increase the distance between two models,

the win rate of the higher-rated model increases. We also see that as ratings go up, the win rates tend to be closer to 50% . In general, these win rates correlate with the general notion of chess Elo ratings, but the magnitudes are not the same. In the standard Elo system, a 200 point difference should correspond to a 75% win rate, but in our models it corresponds to between 51% and 58% win rate. Jacob et al. (2022) showed that higher-rated Maia models trained using standard imitation learning were suboptimal, and claimed that this would lead to weaker play than expected. Our results support this claim, by showing that the difference between (for example) a Maia-1800 and Maia-1600 is smaller than the difference between an 1800-rated human and a 1600-rated human.

## 3.3 CURRICULUM TRAINING

We formally introduce the notion of *curriculum training* as the data-efficient fine-tuning of an existing human-like model, with the dual objectives of improving the model's strength and preserving the human similarity of the original model.

In curriculum training, we start with a human-like model $M$, which we call the *student*. Our goal is to choose a pair of *curriculum* $X$ and *teacher* $Y$ to update the given model. A curriculum is specified by the input distribution $X$. For example, $X$ could be a distribution of chess positions that players of a certain rating have encountered in their games. A teacher is specified by the labeling strategy $Y$, which specifies the labels for each input $x$ drawn from $X$. For example, $Y$ could be the move a particular human would make in the given position, or the move a superhuman engine would make.

After specifying the curriculum and teacher $(X, Y)$, we draw data points $\{(x, y)\}$ from the provided distribution and use them to update the model $M$. Following the curriculum learning approach, we update the model by performing gradient descent using this data. Note that our curriculum training notion extends the standard curriculum learning notion developed by Bengio et al. (2009) to settings with multiple teachers (labeling strategies). To evaluate the performance of curriculum training, we vary the amount of data used to update the model $M$ and measure the strength (match win percentage) and human similarity (move-matching accuracy) of the updated model, as per §3.2.

In this work, we use the domain of chess to evaluate the effectiveness of curriculum training. We pre-trained a set of Maia models to serve as the initial human-like model $M$ and the potential teachers $Y$. We vary the choice of teachers $Y$ to be different ratings of human players or human-like models, or a superhuman chess engine, and examine their impact. We also examine the impact of choosing curriculum $X$ based on the potential mistakes that the student might make, using the assistance of a superhuman chess engine, stronger-rated Maia models, or a human expert.

## 4 EXPERIMENTS

In this section, we examine the effectiveness of curriculum training in the domain of chess. We adopt the Maia-1200 model as our student (i.e., the initial human-like model we wish to improve), and update it using different designs of curriculum $X$ and teacher $Y$, varying both $X$ and $Y$ (§4.1), varying only $Y$ (§4.2), and varying only $X$ (§4.3). We evaluate the effectiveness of curriculum training on our three performance measures: data-efficiency, human similarity, and model strength. We then extend our evaluation of human similarity to cover datasets of higher-rated players, to measure the progress of the student with respect to higher-rated human-like models (§4.4).

### 4.1 EXPERIMENT 1: HIGHER-RATED PLAYER DATA

We first examine the most natural design of the curriculum and teacher pair: updating the initial human-like model Maia-1200 using data from higher-rated human players. Specifically, let $X_R$ be the distribution of positions encountered by $R$-rated players, and $Y_R$ be the moves made by $R$-rated players in these positions. (Incidentally, this is the dataset used to train the Maia-$R$ models.) In this experiment, we set our student $M$ as Maia-1200, and we vary the rating $R$ from 1400 to 3000 and use $(X_R, Y_R)$ as the (curriculum, teacher) to update the student. To investigate the efficiency of this design, we gradually increase the amount of data used to update the student model. For each resulting model (updated using a different amount of data and different $(X_R, Y_R)$), we measure its human similarity (i.e., move matching accuracy) and strength (i.e., match win percentage).

For each design from $R = 1400$ to $3000$, we fine-tune the student for up to 10,000 batches of data drawn from $(X_R, Y_R)$, where a batch is a set of 1024 data points. Due to a lack of data for higher-rated players, the size of the training curriculum is smaller for $R = 2600, 2800$, and $3000$ (with 7300, 900, and 100 batches, respectively). We evaluate win percentage with a match size of 1024 games, and move-matching accuracy using 10 batches. We compute these evaluations every 10 batches between batch-0 and batch-100, and every 100 batches between batch-100 and batch-10,000. We repeat each set of evaluations 10 times, for a total of 10,240 games and 100 batches per evaluation, and report the moving average (with a window size of 50) of these measurements.

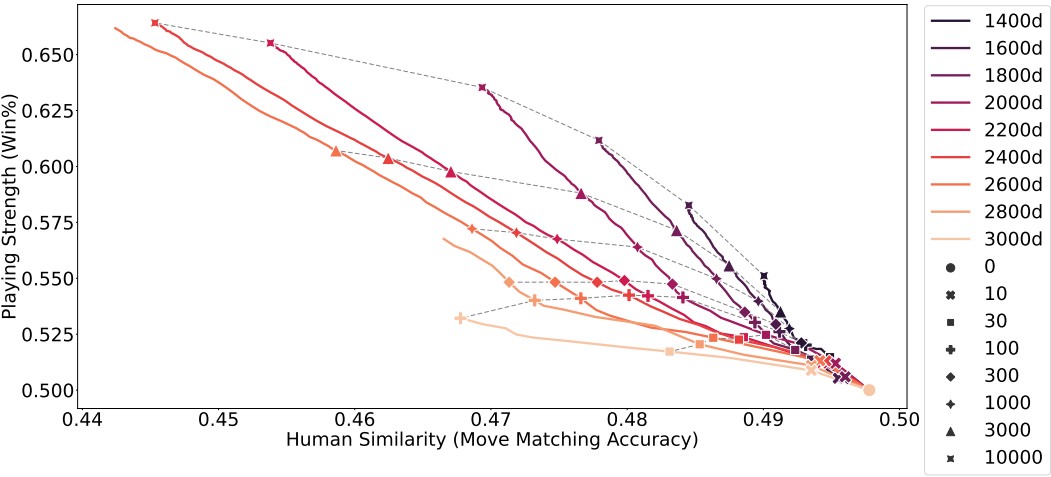

Figure 2: The results of using higher-rated player data to fine-tune a lower-rated student model. Each curve represents the change in human similarity and strength when fine-tuning the Maia-1200 model on increasing amounts of data from players of a particular rating. For each curve, we label the points that are trained using different numbers of batches, from 10 batches to 100,000 batches.

**Experiment results.** The results, as displayed in Figure 2, demonstrate the trade-offs of our three performance metrics for different pairs of curriculum and teacher $(X_R, Y_R)$. First of all, if our goal is purely to increase strength without concerns for human similarity or data efficiency, training using the highest rated curriculum leads to the strongest model.

If our objective is to increase the model's strength while maintaining the model's human similarity above a selected threshold (without concern for data efficiency), using the closest-rated data for fine-tuning leads to the optimal outcome. For instance, if we want to maintain a move-matching accuracy of at least 49%, the optimal strategy is to train on 1400-rated data curriculum for 10,000 batches, which gives us a model with a 55% win probability. We could more efficiently achieve a 55% win probability by training on a 1800-rated curriculum for 1000 batches, but at the cost of a much lower move-matching accuracy. The drawback of using the closest-rated curriculum is that training on low rated data will eventually hit a plateau—not only do we require an order of magnitude more data to achieve the same performance, but the overall gains are limited. Figure 1a suggests that training a 1200-student on 1400-rated data can only achieve 57% win rate, even with an infinite amount of data, while higher-rated curricula easily surpasses this limit.

If our objective is to maximize strength gains with a constraint on data usage, we see an interesting and non-trivial result: at many points during fine-tuning, the optimal curriculum is not the highest-rated or lowest-rated one, but at a point in between. For example, at batch-30, the 2000-rated curriculum is optimal, while at batch-100 and batch-300, the 2400-rated curriculum is optimal. Past this point, it appears that the 2600-rated curriculum is optimal at all batch sizes. However, it would not be surprising if the trend continues: if we had access to the 3000-rated curriculum at higher batch sizes, the high-rated model might perform worse than a 2600-rated or 2800-rated curriculum.

Overall, these results suggest that the strength of a teacher matters significantly when training models. Note that in this experiment, we have simultaneously updated the curriculum $X$ and the teacher $Y$. In order to separately examine the effects of curricula and the effects of teachers on the fine-tuning process, we run two additional experiments to investigate these factors separately.

## 4.2 EXPERIMENT 2: EFFECT OF TEACHERS

To understand the effect of teachers on training, we hold $X$ constant and vary $Y$. For these results, we chose to fix $X$ as the set of board positions that the student was initially trained on. In this case, $X$ would be the distribution of board positions encountered by 1200-rated players.

However, when we fix the input distribution $X$, we are unable to vary $Y$ as in the previous experiment, as we might not have data on which moves, say 1400-rated players, would actually make in our selected $X$ distribution. As an approximate solution, we use Maia models that we have trained to serve as the teacher $Y$. Thus, we teach Maia-1200 using a curriculum where $X$ is the set of positions encountered by 1200-rated humans, and $Y$ are the moves that Maia-1400 would make in these positions. We also introduce an additional possible teacher, Stockfish, the strongest chess engine in the world. This allows us to approximate much higher rated humans, since no models higher than Maia-2600 are available. Using this curriculum design, we ran another experiment with the same training procedure and evaluation methods as Experiment 1.

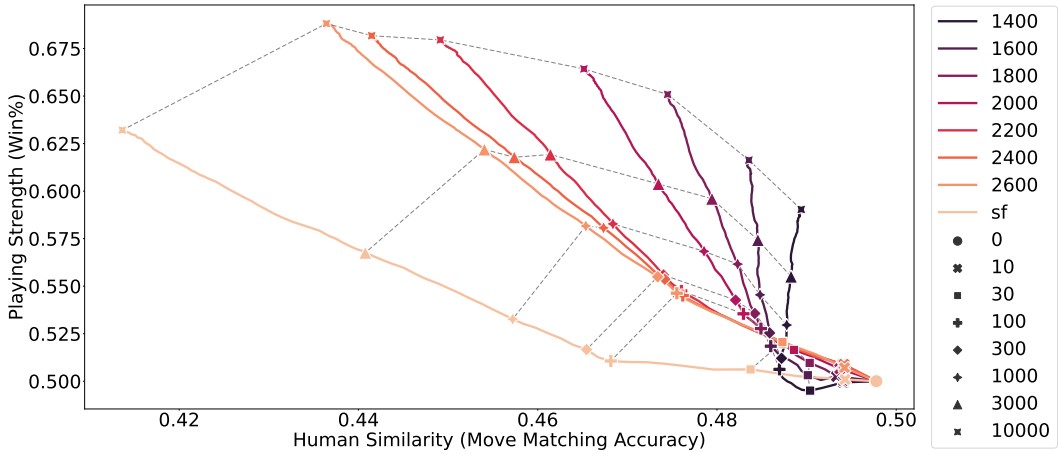

Figure 3: The results of varying the teacher $Y$ to Stockfish (a superhuman chess engine) and Maia models of different ratings, while fixing the input distribution $X$ to the student's positions.

**Experiment results.** The results are displayed in Figure 3. We see that, similar to the earlier experiment, move-matching accuracy is monotonically decreasing with respect to teacher rating. We also see the same non-trivial result that the most data-efficient curriculum is often not the highest or lowest-rated teacher, but somewhere in the middle. In particular, the most efficient teacher is Maia-2200, Maia-2400, or Maia-2600, and these three points tend to be clustered closely together.

Interestingly, we see that Stockfish performs noticeably worse than most of the human-rated models. This could purely be because Stockfish's strength outclasses all humans, so much of the move advice Stockfish teaches is out of reach for the Maia-1200 student. An alternate hypothesis is that Maia-1200 is unable to account for the stylistic differences between a human-like model and an alpha-beta model like Stockfish. It would be an interesting future direction to clarify this question, e.g., by running curricula taught by lower-rated chess engines. Overall, these results confirm our observation in Experiment 1 that the teacher has a significant effect in fine-tuning a human-like model.

## 4.3 EXPERIMENT 3 - EFFECT OF CURRICULA

We now hold the teacher $Y$ constant and vary $X$, which board positions we teach from, to examine the effect of curricula. From the previous experiment, we see that Maia-2200, Maia-2400, and Maia-2600 are the three most efficient teachers for retaining human similarity and improving playing strength, and are all very similar to each other overall. Accordingly, we take the middle of these three, and choose our teacher to be Maia-2400 for the next set of experiments.

One possible class of curricula we could analyze is the distributions of board positions encountered at different levels. However, these distributions do not qualitatively differ very much from one another. Instead, we use specific domain knowledge in chess to develop our own sets of curricula

based on fixing mistakes made by the student. We list three categories of mistakes, and choose one representative curriculum for each category to analyze.

- Mistakes identified by a *superhuman chess engine*. We choose the position of blunders, which is defined as a move which lowers expected win probability by over 15% as predicted by Stockfish.

- Mistakes identified by a *human expert*. We have collaborated with an International Master (IM) on this work. The IM analyzed 80 simulated games played by Maia models at various strengths, and determined that Maia tends to play suboptimally in the opening phase of the game, especially in positions where Maia should "connect the rooks". As this is also a common problem for human chess players, we selected this as our representative curriculum.

- Mistakes identified by a *higher-rated human-like model*. The final category is positions where higher-rated Maia models disagree with the move that the lower-level Maia-student would make. While this is not conclusive evidence of a mistake, it is a possible indication that the lower-level Maia is playing suboptimally. We selected positions where Maia-1200 and Maia-1400 differ, which we denote as 1400-inflection positions.

Using these curricula and the Maia-2400 teacher, we ran another experiment with the same training procedure and evaluation methods as Experiments 1 and 2. For context, we also present a baseline curriculum which is just a random set of positions encountered at the student's rating level.

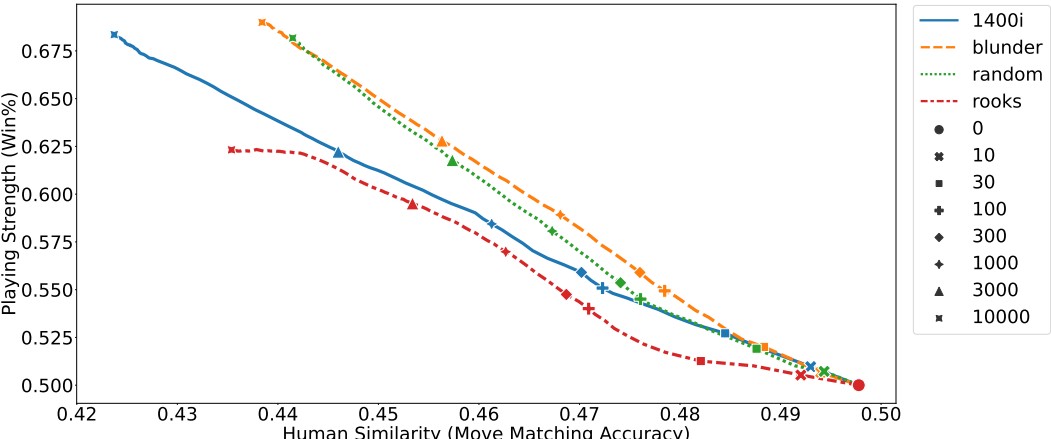

Figure 4: The results of fixing $Y$ to Maia-2400 while varying $X$ to positions identified by a superhuman chess engine (blunder), a human expert (rooks), and a human-like AI (1400-inflection).

**Experiment results.** The results are displayed in figure 4. If our objective is to maximize strength gains with a constraint on the data usage, both the blunder curriculum and the 1400-inflection curriculum are more efficient at teaching the base model than a random curriculum, while the rooks curriculum underperforms in this regard. One hypothesis is that the rooks curriculum (human expert recommendation) is too specific in the positions it tries to teach. While it is true that both human students and Maia models tend to make mistakes in the opening phase of the game, it is likely that these decisions are not game deciding, so it may be less critical to look at these positions. On the other hand, the blunder and 1400-inflection curricula (positions identified by superhuman and human-like AIs) look at mistakes that the model makes across the entire distribution of general board states, but the emphasis on mistakes leads to better outcomes than no emphasis. If our objective is to increase the model strength while ensuring the model maintains human similarity above a selected threshold, the blunder curriculum outperforms random while the 1400-inflection and rooks curriculum do not. However, the difference in performance is small across all curricula.

Overall, our results demonstrate that, among a general set of curriculum we have examined, altering curriculum only brings marginal benefits compared with using random curriculum.

## 4.4 EXPERIMENT 4 - MEASURING HUMAN SIMILARITY

In the previous set of experiments, we evaluated human similarity by using the 1200-rated human dataset. Because the student model was originally trained and evaluated on this dataset, it makes sense to use this same dataset as a way to measure how much the updated model has deviated from the original model. In domains where human data is not labeled according to individual skill, this is indeed the only option for evaluation, and thus our previous approach should generalize to domains with human-like models trained in this way.

However, thanks to the rating metadata included in the human chess dataset, we can go one step further in our analysis. After a model undergoes curriculum training, the model will naturally degrade in its initial performance, because we are teaching it to play differently. However, the updated model might in fact play more like a higher-rated model because we have taught it to play better.

To investigate this, we took the 11 models trained in Experiments 2 and 3 after learning the full 10,000-batch curriculum, and analyzed their move-matching accuracy against higher-rated datasets, to see if the updated models play like higher-rated humans.

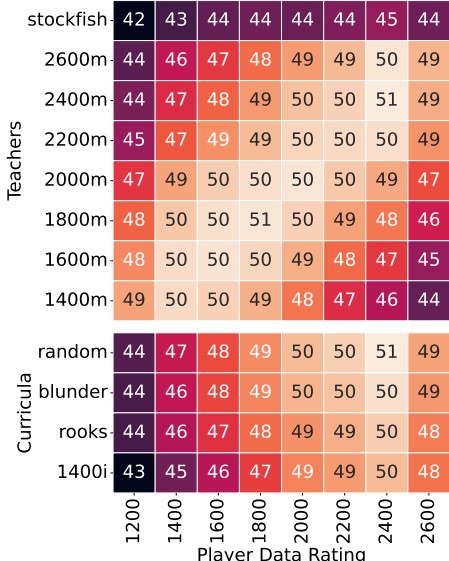

Figure 5: The results of analyzing how similar curriculum-updated models are to human datasets at a range of skill levels.

**Experiment results.** The results are displayed in Figure 5. Overall, we find that the updated models are more similar to higher-rated datasets than the original dataset. Looking at the models from Experiment 2, where we varied the model's teacher, there is a very strong correlation between the rating of the teacher and the most similar dataset of the updated model. For instance, the model taught by Maia-1600 is most similar to the 1600-rated dataset, while the model taught by Maia-2400 is most similar to the 2400-rated dataset. Interestingly, each of the models taught using targeted curricula and Maia-2400 teacher all perform most similar to the 2400-dataset. This provides more credence to our earlier observation that the teacher is much more important than the curriculum.

## 5 CONCLUSION AND DISCUSSION

Despite a growing effort to develop human-like models that capture human behavior at different granularities (such as skill levels), surprisingly little attention has been paid to how these models can be trained or improved. In this paper, we propose a generalized framework for improving a human-like model while maintaining its human similarity, using a novel extension of curriculum learning which considers both the curriculum data to train on, and the teacher used to label this data. We measure performance as a trade-off between the improved model's strength, its human similarity, and the data efficiency of the curriculum.

We evaluate this framework in the domain of chess through a series of experiments. At a high level, we find that the choice of teacher is critical for maximizing the efficiency of strength gains. In many cases, the best teacher is *not* the strongest teacher. Notably, Stockfish (the world's strongest chess engine) is a worse teacher than even a low-rated human-like model. Our results also suggest that the choice of curriculum is less important than the choice of teacher. Although we did find evidence that efficient curriculum design is possible, it remains an open question how best to select a curriculum for training human-like models in chess.

It is important to note that any insights we gain from training human-like models do not necessarily apply to training real humans. These are, after all, neural network models, and not human brains. Nevertheless, the results in §4.4 suggest that our curriculum training methodology is able to improve human-like models in a way that ensures human similarity at their improved level of play, on a path similar to human learning. This could provide insights into training real human students and is a topic we plan to explore in future work.

## ETHICS STATEMENT

This work develops a methodology for improving the strength of a given human-like model, while maintaining its human-like property. The human-like models we create are trained on aggregate human data grouped by coarse skill levels, similar to the models trained by McIlroy-Young et al. (2020) (though our strongest human-like models surpass theirs in target Elo rating). We do not train or make use of any individualized models in this work.

While there are many potential societal benefits to creating human-like models (as discussed in the introduction), there are also potential ethical concerns associated with creating, deploying, or using such models, especially when the models capture individual human behavior. McIlroy et al. refer to such models as "mimetic" and explore these ethical concerns in McIlroy-Young et al. (2022a). For example, a human-like model of a target individual may leak private information about that individual, may allow others to prepare or plan for future interactions with that individual (by practicing with the model), and my even allow the individual to multiply their own output and interactions (by using copies of the model as "proxies" of themselves). Moreover, the presence of such models could introduce new forms of human-AI interactions, raising new questions about how humans perceive, trust, and evaluate these AI systems. In general, we would need more careful domain-specific consideration in order to address these questions.

In this work, we train human-like models using a publicly available dataset from Lichess.org. For future research that would like to extend our methodology to other domains and/or examine its impact on real humans, proper review and approval from an IRB would be required.

## REPRODUCIBLILITY STATEMENT

All the data needed to reproduce our results is available at https://database.lichess.org. Our anonymized source code is available at https://anonymous.4open.science/r/chesstrainer-DBBE. For guidance on how to use this repository to reproduce our code, we have provided a README.txt file. Details on how the models were trained is available in the Appendix.

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

## A    APPENDIX

## A.1 MAIA MODEL TRAINING

**Data Pipeline**

The raw data used for this work was downloaded from http://database.lichess.org. The specific files we used were from the months 2021-06 to 2021-03 for the training data, and 2022-04 for the test data. We also use the 2022-04 data for evaluating the result of our curriculum experiments.

While McIlroy-Young et al. (2020) published code for preprocessing the data, we decided to write our own code from scratch using a tensorflow dataset pipeline. Using this custom pipeline, we were able to store the entire preprocessed dataset using only 231GB, while the original pipeline would have used over 3TB. In addition, this training pipline massively sped up training time—training a full model using the existing published code would take around 30 hours on an NVIDIA V100, while the new pipeline's efficient handling of IO reading and GPU allocation means we can train the same model in under 9 hours using the same GPU. We have made this code available, and hope that these improvements make the training process more accessible to the public.

**Model Training**

The majority of the details of our model are identical to the models used in McIlroy-Young et al. (2020). We use the same training size (400,000 batches), optimizer, and learning rate schedule. The architecture of our neural network is almost identical, except that we remove the value head, resulting in purely supervised learning.

Nevertheless, there are a few differences between the prior models and our work. First, we restrict the training and test datasets to exclusively use blitz data (i.e., games between 3 and 8 minutes), while the prior work included all games above 3 minutes. Since games with 3-8, 8-20, and above 20 minutes are scored using different rating pools, conflating these different systems could lead to a less fine-grained understanding of human behavior, since the model may get confused by conflated rating levels.

We also discard game history while training models. While this sacrifices the prediction accuracy slightly (by 3%), it aligns better with our goal of designing a curriculum, since a curriculum may span positions from various games (at varying points in each game), which may not preserve the context of a game's history. We also expand the models to incorporate higher-rated styles. The prior work trained models rated between 1100 and 1900, while we added additional higher-rating models until the rating of 2600 to facilitate a richer discussion of curriculum design and model improvement. A lack of data above 2600 prevents us from training Maia models at higher levels.

