# OpenReview forum: "Improving the Strength of Human-Like Models in Chess"
_ICLR.cc/2023/Conference — Submitted to ICLR 2023_

### Official Review · Reviewer_9kEs · 2022-10-14

**Confidence:** 3
**Correctness:** 2
**Technical Novelty And Significance:** 1
**Empirical Novelty And Significance:** 2
**Recommendation:** 3

**Clarity, Quality, Novelty And Reproducibility:**

The paper is well written, but some clarity issues need to be resolved:
- "behavioral data used to train the original model" seems to relate to the Maia-1200 model ?
- Without knowledge of Bengio et al., the approach is not clear. Especially, because only a one-step "Curriculum Learning" seems to be used.
- Figures 2,3 is a bit hard to decipher, because of the high amount of information contained. Maybe another representation form (table, multiple figures, ...) can remedy the issue.
- Please state the ELO (>2400) of an International Master. This relevant to determine the quality of the "rook" subset.

Reproducibility is assumed, given the source code is available.

**Strength And Weaknesses:**

The authors offer a quite extensive evaluation of the proposed approach as well as the basic models used. This evaluation includes a consideration of differences between the used positions and the source (teacher) of the proposed move. Therefore, the evaluation is beyond what is usually expected. However, the used definition of the "human similarity" metric is problematic. For one, expecting an exact matching of ground truth moves is very strict and a similarity metric should be preferred. Although, this is likely not a trivial task, because similarity to human play can potentially be only evaluated by human experts. However, the phrase "behavioral data used to train the original model" suggested, that in most experiments, only moves from the Maia-1200 set are considered. (Also see clarity) In case this understanding is correct, the metric only evaluates deviation from novice play - not from human play. Therefore, the correlation between increase in playing strength and decrease of similarity becomes obvious. A potential resolution could be comparing to any human play (independent of the playing strength or up to the teachers playing strength). This should be an approximate if any human would play like the engine.
Another problem is, that it seems, that learning approach has not converged, after using all batches once (Fig 2,3). Without convergence, the used learning rate becomes an interesting question, because it implicitly defines the trade-off between initial weights and new data.
Furthermore, Section 3.3. is a bit lacking - considering it is the introduction of the main approach. (See clarity)

**Summary Of The Paper:**

The authors analyze an approach to improve the playing strength of human-imitating chess engines, while trying to maintain the human-like play. The basic method is to use transfer learning (one-step curriculum learning) by retraining an existing human-like model with data from better players or a conventional chess engine. The evaluation focuses on the effect of the difference in playing-strength between initial model and curriculum data, in relation to the human similarity. Additional, a subset approach is evaluated, using only specific chess positions or a random subset.

**Summary Of The Review:**

The paper does not propose a novel method, but performs a detailed analysis of an existing approach applied to the domain of chess. This is a niche topic, but substantial evaluations and conclusion may still justify acceptance. However, in case the reviewers assumption concerning the Maia-1200 set and the human similarity metric is correct, all conclusions become questionable. Therefore, the evaluations are not sufficient for acceptance. Furthermore, the stated clarity issues should be resolved.

---

### Official Review · Reviewer_xpsR · 2022-10-23

**Confidence:** 4
**Correctness:** 3
**Technical Novelty And Significance:** 1
**Empirical Novelty And Significance:** 1
**Recommendation:** 3

**Clarity, Quality, Novelty And Reproducibility:**

As expanded above, I find the work to be of high clarity and little novelty. I suspect the results are easily reproducible, most of them also in other domains.

**Details Of Ethics Concerns:**

I selected "Yes, potentially harmful insights", as the paper's ethics statement doesn't address the fact that improving human-like models will make it harder to detect cheaters (which is a growing concern in the chess community).

At the same time, I chose "NO", as in my mind the contributions of the paper are not significant enough to directly lead to an improvement in this space.

**Strength And Weaknesses:**

## Strengths
The writing of the paper is exemplary, making it relatively easy to understand. I particularly liked the sec. 1 motivation behind the need for human-like models, even though they may have a worse performance than the models not taking this into account.

## Weaknesses
I find the evaluation of the human likeness of the models to be performed on the original datasets (of the original training, not the finetuning) unacceptable. Doing this makes the metric to count both "unhuman" and "human, but of higher-rating" as unhuman behavior, which doesn't follow the motivation of improving the models while keeping them human. While the sec. 4.4 shows partial results on the target rating, I find presentation of the original rating results as confusing, and the sec. 4.4 to not be detailed enough to override this assessment.

If the proper train/test split was not used, the results may suffer from overfitting (ie. the reported human likeness would not constitute a good approximation of a real move-matching accuracy on new positions).

The presented results provide no new insights for me, and are predictable:

1. As human likeness measures accuracy of reproducing the original data, it drops when finetuning on different data [4]. The higher the rating gap (=> bigger domain shift), the bigger the catastrophic forgetting.
2. As we train on the data with humans of a better rating, the performance (assuming enough data) improves, as the model gets to reproduce the better moves.
3. The initial speed of improvement is lower for datasets that are more different (ie. higher rating) from the original, due to less common features being reusable in the new domain. This implies that, for a fixed sample budget, to get the best final performance there will be a tradeoff between low rating (=> the domain is close and there are a lot of reusable features), and high rating (=> overall better moves to reproduce). Authors also provide a correct intuition in the paper behind this: "After a model undergoes curriculum training, [it] will naturally degrade in its initial performance, because we are teaching it to play differently".

I don't find the results in Experiment 3 very informative, as it's hard to understand the difference between the handcrafted data distributions (is the score lower because the data is of less variance, or coming from different-rating source, or the chosen positions don't cover the space of positions well enough to improve the model, etc.).

The benefits of introducing the curriculum training framework are not clear to me: the concrete method for updating the network (finetuning) is prior work, as is generating the actions from previous trajectories or other models. The paper makes no claims about any of the instantiation of the framework (using trajectories or other models' actions, or changing the states only) as being empirically better then the others, nor does it compare to a natural baseline of training on the new dataset from scratch.

## Actionable feedback

1. Present the results in the experimental section on the target domain (ie. it's human if it's close to what the human of a given strength would play)
2. Clarify whether a train/test split was used in the paper and if not, provide the results where the evaluation data is separate from the training one.
3. Provide a clearer value for an ICLR reader: what it is actually supposed to learn from the paper? The reader would be aware of the techniques for improving the models from the literature, is one of the methods better than the others, in what situations? Which of the claims are only applicable in the domain of human-like models or domain of chess? Is there anything else to take home?

[4]: https://en.wikipedia.org/wiki/Catastrophic_interference

**Summary Of The Paper:**

The topic of the paper is finetuning models trained to play chess like humans (defined as: making the same moves as in a dataset of human games) to improve its performance (ie. chess rating) without losing the human-like behavior. Authors introduce a framework called curriculum training, and "use it" to update the model trained on human games with a certain rating to play like humans of a higher rating.

Curriculum learning consists of an idea that the game positions (which authors refer to as curriculum, and RL terminology would call states) can be chosen independently from the strategy of choosing target moves (which RL terminology would call action/policy and authors refer to the teacher or labeling strategy).

Authors choose various settings where the models trained on games of humans of a given strength are finetuned in a supervised way to reproduce moves of higher-ranked humans:

1. Where both states and actions come from the human recorded data (Experiment 1), in a way similar to Behavioral Cloning / Imitation Learning [1]
2. Where the states come from the human data, but the actions are coming from a model trained to imitate human moves (Experiments 2, 3), which has similarity to distillation [2, 3]

In these experiments, authors study the effect of increasing the amount of data and the strength of the humans in the target dataset / model on two metrics:
1. accuracy of the updated model to predict the moves of the humans of the dataset the was originally trained on (with lower score) (human likeness)
2. performance of the updated model in games with similar models

In Experiment 4, authors shortly discuss human likeness, as measured on the target dataset.

The phrasing of the paper doesn't make it clear whether the data used to measure the human likeness is the same as the one the models have been trained on, or whether a proper train/test split was applied.

[1]: Abbeel et al. [Apprenticeship Learning via Inverse Reinforcement Learning](https://ai.stanford.edu/~ang/papers/icml04-apprentice.pdf)

[2]: Hinton et al. [Distilling the knowledge in a neural network](https://arxiv.org/abs/1503.02531)

[3] Furlanello et al. [Born again neural networks](https://arxiv.org/abs/1805.04770)

**Summary Of The Review:**

I find the results of the paper presented in a compelling way, but I don't find them interesting enough for the ICLR community. I think that the methods employed by a typical reader of ICLR papers if they wanted to improve a model while keeping it human-like would be the same before and after reading the paper, nor do I see any other benefits they would gain from the lecture.

Nits:

1. Ethics statement: "and m*a*y even allow the individual"
2. In sec. 4.3 authors write "from the previous experiment, we see that Maia-2200, 2400, and 2600 are the three most efficient teachers for retaining human similarity and improving playing strength." However, based on the curves, it seems to me that there is no clear winner there, and the 1400 is offering a better "human similarity".
3. I suspect that the result of "training on a curriculum of errors to provide a marginal benefit over training on random positions" is highly domain-specific, and may not extend beyond the particular datasets tested.

---

### Official Review · Reviewer_aYkk · 2022-10-25

**Confidence:** 5
**Correctness:** 3
**Technical Novelty And Significance:** 1
**Empirical Novelty And Significance:** 2
**Recommendation:** 3

**Clarity, Quality, Novelty And Reproducibility:**

* Clarity and quality: The is well-written but there are some issues that have been addressed in the previous “Strength And Weaknesses”.
* Novelty: The originality of this paper is poor and the experiments do not have new contributions.
* Reproducibility: The authors provide the source code so the reproducibility is good.


**Details Of Ethics Concerns:**

The authors have addressed the ethical concerns in the main paper. It might be careful for the trained models to expose the tendency of real humans while playing chess. Especially for Maia-3000d, which has smaller data, it is easier to target the player that the model has mimicked.

**Strength And Weaknesses:**

Strength:
This paper studies curriculum learning and human-like models in the widely-studied domain of Chess. They provide detailed experiments to find the relation between strength and similarity.

Weaknesses:
However, there are no new techniques in the methodology, and the experiment results are not surprising. To be more specific, they used transferring learning to fine-tune a 1200m model with datasets of different ratings. It is very obvious that the playing strengths will increase and the human similarity will decrease when comparing the fine-tuned model with the original one. Besides, in subsection 4-4, the author said “the model taught by Maia-1600 is most similar to the 1600-rated dataset, while the model taught by Maia-2400 is most similar to the 2400-rated dataset”. This conclusion does not give any new findings. Overall, there are no new findings and the novelty is poor.

Other comments:
* Fig. 1 and Fig. 5 only provide the decimal numbers, while the readers might be interested in the actual difference in smaller grains.
* In experiment 4.1, the author said “at least 49% … the optimal strategy is to train on 1400-rated data curriculum for 10,000 batches … with a 55% win probability”, and “a 55% win probability by training on a 1800-rated curriculum for 1000 batches but at the cost of a much lower move-matching accuracy.” But according to Fig. 2, the move-matching accuracy is about 48.6% for 1800-rated curriculum of 1000 batches, which is not “much lower” as stated in the context. The results seem to have not much difference when comparing different settings.
* The author should explain how a move is generated by the trained model, is it by MCTS or just by the move with the maximum policy?
* The author claims to propose a framework for curriculum learning, but according to the contents of the paper, this seems exaggerated.
* In subsection 4.3, the author should provide the total number of positions for the three categories of mistakes. Especially for human experts, the 80 simulated games seem to be much smaller than the other two categories which can be generated by programs.
* In Fig. 1 and Fig. 5, what does the color mean?
* On page 5, “50% .” -> “50%.”
* In Data Pipeline section of the appendix, “from 2021-06 to 2021-03”, is it typo for “2021-06 to 2022-03”?


**Summary Of The Paper:**

This paper extends the concept of curriculum learning and investigates the question of teaching an existing human-like model to be stronger using a data-efficient curriculum, while maintaining the model’s human similarity. The authors then applied the method to chess and experiments show that the choice of teacher has a strong impact on both playing strength and human similarity. Based on their experiments, they claim that strengthened models maintain human similarity on datasets corresponding to their strengthened level of play.

**Summary Of The Review:**

Overall, the paper has poor novelty and the experimental results seem too obvious and have no new findings. Please refer to “Strength And Weaknesses” for more details.

---

### Official Review · Reviewer_XUrX · 2022-10-25

**Confidence:** 3
**Clarity, Quality, Novelty And Reproducibility:** See main review.
**Correctness:** 2
**Technical Novelty And Significance:** 3
**Empirical Novelty And Significance:** 2
**Recommendation:** 3

**Strength And Weaknesses:**

Strengths

- **************Clarity**************: The paper is well written and easy to understand, and the sections naturally follow from one to the other.
- **********Novelty:********** The paper is relatively novel in its approach towards understanding curriculum learning and building human-like models. While the zone of proximal development (ZPD: problems with medium difficulty are the most useful) have been studied by measuring problem difficulty, the paper looks at teacher/annotator proficiency.
- The analysis in the paper is thorough, and the graphs and methods are easy to understand.
- The insight of accounting for labeller/ teacher proficiency is quite powerful.

Weakness

- The interpretation of the result (most data-efficient curriculum is often not the highest or lowest-rated teacher, but somewhere in the middle) in Experiment 2 is not well supported. The authors themselves suggest the alternate hypothesis that seems more likely: Maia- 1200 cannot account for the stylistic differences between a human-like model and an alpha-beta model like Stockfish. The alternate hypothesis seems likelier as the most similar stylistic teachers with the highest ratings are the best teachers. Without an experiment to test this, one of the central claims of the paper is not on firm grounds.
- I am unsure how the results in this paper might extend to a domain outside of chess: either for teaching people or for building stronger human-like models. The discussion section is lacking here, with just one application domain in the paper! For example, the claim that “that the teacher is much more important than the curriculum.” might not extend to other domains.
- Experiments 2,3 and Figures 2 and 3 show human similarity based on the games of 1200 rated players. When improving the strength of a model, one might want to see a resemblance closer to the target rating rather than the base rating. So, if I wish to have a certain similarity with a 1600-rated human, would it be better to train using a 1600 agent or a 1400 agent or an 1800 agent? Using matching accuracy from 1200-rated games does not seem right (this is also seen in experiment 4).

Minor

- The caption or the graph in figure 5 does not mention the metric shown.

**Summary Of The Paper:**

The paper proposes a curriculum learning paradigm to improve the strength of human-like chess-playing models. The paper studies how the occurrence of problems at a level and the labelling strategy affect the learning process. Through various experiments and analysis of the strength of the teacher and the selection of problems, the authors find that 1) the strength of the teacher labelling the problems matters much more than the curricula of problems 2) The most efficient teacher to improve performance is not the most or the least expert but an intermediate one.

**Summary Of The Review:**

The paper proposes a new dimension of axis for exploring curriculum design, teacher expertise. This is quite an interesting insight but the paper in its current form does not sufficiently support its claims.

---

### Decision · Program_Chairs · 2023-01-20

**Decision:**

Reject

**Justification For Why Not Higher Score:**

The main concern is that there are no new techniques in the methodology, and the experiment results are not teaching us anything new.

**Justification For Why Not Lower Score:**

NA

**Metareview: Summary, Strengths And Weaknesses:**

There was strong agreement amongst the reviewer that the paper currently is not ready for publication at a major conference.